# Selected Properties of the Surface Layer of C45 Steel Parts Subjected to Laser Cutting and Ball Burnishing

**DOI:** 10.3390/ma13153429

**Published:** 2020-08-04

**Authors:** Agnieszka Skoczylas, Kazimierz Zaleski

**Affiliations:** Department of Production Engineering, Faculty of Mechanical Engineering, Lublin University of Technology, 36 Nadbystrzycka, 20-618 Lublin, Poland; k.zaleski@pollub.pl

**Keywords:** laser cutting, ball burnishing surface roughness, microstructure, microhardness, residual stress

## Abstract

In this article, we report the results of experimental studies on the impact of ball burnishing parameters on the roughness, microstructure and microhardness of the surface layer of laser-cut C45 steel parts. We also analysed the distribution of residual stresses generated in the surface layer of these parts. Laser-cut parts often require finishing to improve the quality of their surface. The tests performed in this study were aimed at assessing whether ball burnishing could be used as a finishing operation for parts of this type. Ball burnishing tests were performed on an FV-580a vertical machining centre using a mechanically controlled burnishing tool. The following parameters were varied during the ball burnishing tests: burnishing force *F_n_*, path interval *f_w_* and the diameter of the burnishing ball *d_n_*. Ball burnishing of laser-cut C45 steel parts reduced the surface roughness parameters *Sa* and *Sz* by up to 60% in relation to the values obtained after laser cutting. Finish machining also led to the reorganization of the geometric structure of the surface, resulting in an increase in the absolute value of skewness *Ssk*. This was accompanied by an increment in microhardness (maximum microhardness increment was Δ*HV* = 95 HV0.05, and the thickness of the hardened layer was *g_h_* = 40 µm) and formation of compressive residual stresses in the surface layer.

## 1. Introduction

Ball burnishing has been successfully used in the engineering industry for several decades now. It allows one to obtain hardened surfaces with low roughness, improved resistance to abrasive wear and increased fatigue strength [1,2,3]. This technology is effectively used as a finishing operation in the production of such parts as dies, moulds, shafts, compressor turbine blades, or rear suspension control arms [2].

Burnishing is very often integrated with machining, especially when axially symmetrical elements are machined using a lathe [4], a drilling or a boring machine. When burnishing axially symmetrical parts, the burnishing tool runs parallel or perpendicular to the marks left by the machining tool (a drill, a boring bar, or a turning tool) [1,5,6]. Recently, ball burnishing has also been successfully performed in a single operation in combination with milling on multi-axis CNC machining centres, using commercially available tools [7,8,9].

Ball burnishing is an environmentallyfriendly finishing process. No chips, sparks and dust are generated during the process, and the use of coolant and lubricants is kept to a minimum. This means that ball burnishing can replace traditional finishing operations used in the production of machine parts, such as grinding or lapping [1,8]. Ball burnishing is an effective method of machining parts made of aluminium alloys [8], magnesium alloys [10], ductile iron [11], titanium alloys [12], and steel with a hardness up to 45 HRC [2,3,13] and with increased hardness [1,4,5,14].

During ball burnishing, the surface roughness profile is modified, and asperity summits are flattened and deformed. Ball burnishing of steel with increased hardness can reduce the surface roughness parameters by about 40% to 90% in relation to the initial value [5,14,15]. In the case of light alloys, it is possible to obtain a surface roughness *Sa* of 0.02 µm, which corresponds to the quality of a polished surface [8]. Modification of the surface roughness profile also translates into an increase in the bearing ratio. Linear, or even degressive, bearing area curves obtained in a pre-machining operation change into progressive ones. In one study, ball burnishing of aluminium alloy EN AW-AlCu4MgSi (A) produced a surface with 3D roughness parameters *Ssk* = −0.27 and *Spk* = 0.02 µm [8]. In another experiment, burnishing of 41Cr4 steel caused a (maximum) 20-fold decrease in the *Spk* parameter [5], which may be indicative of an increase in abrasive wear resistance. The values of surface roughness parameters depend on the pressing force [8,10], burnishing speed [5,15], type of pre-machining operation [1,5], burnishing feed [8,9,15], and burnishing strategy [14].

As an effect of the action of the smooth burnishing tool on the workpiece, the material is deformed and thermally plasticized; this mechanism depends on the machining environment [16]. This leads to microstructural changes. Grains are lengthened, and the microstructural cohesion of the material is disrupted [17]. Changes in the hardness and microstructure of ball-burnished material usually occur in the layer of the material located close to the treated surface [4,17]. However, the increase in the hardness of the surface layer leads to the deformation of the material in the deeper-lying layers the workpiece [18]. Changes in the hardness of the surface layer can be determined in destructive tests, by measuring the microhardness of metallographic specimens, and in non-destructive tests, using annihilation techniques [19]. The increase in hardness depends on the type of workpiece, machining conditions [4,17,20], and the type of machining environment [21].

Another very important effect of ball burnishing is the formation of compressive residual stresses in the surface layer. The resulting residual stresses reside at (1.3–1.8 times) greater depths than the depth of cold work. During ball burnishing of axially symmetrical parts, deformation and residual stresses have greater values in the axial direction. Residual stresses measured in the circumferential direction represent from 30% to 45% of the value of axial stresses [18]. Similar results were obtained in [17]. The compressive residual stresses generated in the surface layer of the workpiece as a result of burnishing lead to an increase in resistance to fatigue wear [22].

The phenomena that occur during ball burnishing result in the generation of mechanical energy, part of which is released in the form of heat, and the rest is stored in the material, causing an increase in its surface energy [23]. In addition, the geometric structure of the surface formed as a result of burnishing affects the energy state of the workpiece, leading to an increase in its adsorption activity and chemical activity [24,25].

Today, there exist technological solutions that combine ball burnishing with other technologies, e.g., laser-assisted burnishing (LAB) [26] or vibration-assisted ball burnishing (VABB) [27,28]. Ball burnishing can also be integrated into a single operation with laser alloying [29]. The simultaneous use of a laser beam and a ball burnishing tool allows one to temporarily soften the material locally and thus achieve fuller plastic deformation [26].

The laser beam has been used as a tool in the machine industry since the second half of the 20th century. Lasers are used in welding, marking, engraving, alloying, and cutting.

Laser cutting is used in the automotive, chemical, marine and aviation industries [30]. This technology can be used to separate most engineering materials [31,32,33]. The advantages of laser cutting include a high cutting speed, a low surface roughness, a satisfactory dimensional accuracy, a high level of automation, and a wide spectrum of applications [31]. Of course, this technology is not without flaws. Some of the downsides of laser cutting include restrictions on the thickness of cut materials, which depends on the type of material being cut, the occurrence of unwanted tensile stresses, and changes in the microstructure which produce changes in the mechanical properties of structural elements [34,35,36]. The coherent laser beam may also generate imperfections on the faces and edges of the workpiece. Imperfections, according to [37] are irregularities or deviations from the specified shape or location of cut. As a result of hydrodynamic flow of molten material, gas flow fluctuation and oscillation of the laser head, a characteristic drag line pattern forms on laser-cut surfaces [38,39]. The drag lines have different heights and are spaced at different distances, which means that different areas of the surface of the cut part will have a different quality and shape. There are two characteristic zones: the laser beam entrance area and the laser beam exit area. These zones are separated from each other by the so-called boundary layer separation (BLS) [40].

Laser cutting may damage the edges of the cut part, which then form a heat-affected zone (HAZ) [34]. The HAZ is a thin layer of material which, locally, especially at the edges and corners, has a high hardness. The thickness of the HAZ depends on laser power, the type of gas used for cutting and the pressure of this gas [34,39].

The high temperature generated in the cutting zone not only leads to the formation of HAZ, but also causes thermal stress. Heating of the workpiece to the melting temperature followed by rapid cooling with the assist gas leads to material shrinkage and phase changes. The workpiece is cooled at a variable rate, which, in combination with the dynamics of the phase changes, leads to the formation of tensile stresses. The values and the depth of penetration of the residual stresses depend on the mechanical and thermal properties of the material being cut [34,41].

The imperfections formed during laser cutting on the faces and edges of a part make it difficult or impossible to use the part in the next production stage. It is also difficult to apply paint coatings to those surfaces due to flaking. The imperfections on the faces and edges of laser-cut parts, the presence of drops of solidified material at their bottom edges, and unfavourable properties of the surface layer call for finishing. Flash can be removed from the bottom edge of a workpiece using hand files, deburring machines, wire brushes and belt sanders. The geometric structure of laser-cut surfaces and their properties can be improved by using centrifugal shot peening [42], smoothing with diamond tools [43], or mechanical grinding [44].

Centrifugal shot peening of laser-cut parts [42] leads to the reorganization of the geometric structure of their surface, endowing it with favourable features. Parts finished by centrifugal shot peening had over four times lower surface roughness parameters and a 14 times higher material ratio. The microstructure of the surface layer was reorganized, which resulted in an increase in microhardness (by a maximum of 16%) and formation of compressive residual stresses. In addition, as the shot impacted the processed surface, combustion products were “blasted” or “sheared” off it.

Smoothing of a laser-cut surface with a diamond tool allows one to obtain a several-fold reduction in surface roughness and leads to structural and phase changes (transformation of residual austenite into martensite). At the same time, the surface layer is hardened mechanically. During finishing with a diamond tool, sliding friction occurs between the tool and the surface of the workpiece. This leads to an increase in the temperature of the treated area, which causes phase changes, and ultimately an increase in the hardness of the part in the edge zone, relative to its hardness after laser cutting. An adverse effect of smoothing with a diamond tool is the appearance of discontinuities and microcracks on the processed surface [43].

A laser-cut surface subjected to mechanical grinding [44] is characterized by a lower roughness and a lower thickness of the surface layer. Grinding introduces into the workpiece a state of tensile stress, which promotes cracking of the surface. The resultant defects are in the place of stress concentration and reduce the bearing capacity of the processed surface.

A review of the literature shows that laser-cut surfaces require finishing. The methods used so far in finishing laser-cut parts produce surface defects [43,44], generate tensile stress in the surface layer [44], and often require the use of expensive special tools [42]. The beneficial effect of ball burnishing on the state of the surface layer of parts with a hardness above 45 HRC is well known, but there are no studies showing that this method could be applied in the processing of laser-cut parts. It can be supposed that ball burnishing of laser-cut surfaces should have a positive effect on their properties. To explore this possibility, we performed experiments in which we assessed the impact of ball burnishing on the properties of the surface layer of laser-cut C45 steel parts.

## 2. Materials and Methods

The tests were conducted using 5 mm × 8 mm × 100 mm specimens of non-alloy C45 steel (R_m_ = 740 MPa, R_e_= 430 MPa, hardness = 250 HB). Steel of this grade is successfully used in the production of medium-load machine and equipment parts, such as spindles, shafts, axles, and unhardened gears. The specimens used in the experiment were cut with a LASER Amada 3000 W laser cutter from Amada America Inc. (Haan, Germany) set to the following technological cutting parameters: cutting speed *v* = 1150 mm/min, power: 2.15 kW, frequency *f_Hz_* = 1280 Hz, assist gas pressure 0.06 MPa, laser beam focus position +13 mm. Oxygen was used as the assist gas. Figure 1 shows the experimental design, which includes a set of variable, constant and output (indirect) factors.

In the experiments, laser-cut surfaces were finished by ball burnishing (Figure 2). During burnishing, a ball-shaped burnishing tool rolls without slipping against the surface being processed, deforming the asperity summits produced in the pre-machining operation (Figure 2b). Ball burnishing is used to smooth and harden the surface, reduce shape defects, carve lubrication microgrooves and increase fatigue strength. In some cases, ball burnishing can replace grinding and lapping operations.

Laser-cut C45 steel specimens were ball-burnished on an FV-580a vertical machining centre (MOC Mechanicy Pruszków, Pruszków, Poland), using a special burnishing tool. The mechanically controlled burnishing tool comprises a casing enclosing a helical spring, which allows to press the working unit attached to the spring against the workpiece. The burnishing machine is equipped with a replaceable tip with a *d_n_* diameter ball made of Si_3_N_4_ silicon nitride. The ball exhibits low adhesion to steel.

A schematic of the ball burnishing operation is shown in Figure 2a. The burnishing tool (1) is secured in the spindle; the machine table with the vice moves linearly at speed *v*, causing the burnishing ball, which is part of the tool mounted in the non-rotating milling spindle, to roll against the workpiece. After each pass, the milling table does a transverse motion that corresponds to path interval *f_w_*. During ball burnishing, flat specimens (2) were fixed in a holder (3), which was clamped between the jaws of a vice (4) mounted on the table of the machine tool.

The ball burnishing operation was carried out using Mobile Cut cooling-lubricating liquid (Mobilcut 100, Mobile, Bavaria, Germany). The parameters of the operation are presented in Table 1. The parameters were selected based on preliminary research. The sample size was established as *n* = 7. Figure 3 presents the surface after laser cutting and laser cutting and ball burnishing.

Three-dimensional topography measurements were performed using a Hommel-Etamic T8000RC 120–140 mechanical roughness tester (Jenoptik, Villingen-Schwenningen, Germany). The scanned area was 6 mm × 6 mm. The TKU300 measuring tip travelled along measuring paths which were perpendicular to the traces of the asperities produced by machining. Hommel Map Basic software(Version 6.2, Jenoptik, Villingen-Schwenningen, Germany) was used to determine 3D surface roughness parameters.

The microstructure of laser-cut specimens and laser-cut and burnished specimens was assessed using a Nikon Eclipse MA 100 metallographic microscope (Nikon, Castle Donington, UK) and a PhenonProX scanning microscope (Thermo Fisher Scientific, Massachusetts, Waltham, MA, USA).

Microhardness measurements were made using the Vickers method in accordance with EN ISO 6507-1: 2018, using metallographic specimens prepared in a standard manner. An LM 700at microhardness tester (Leco, St. Joseph, MI, USA) was used at indenter loads of 10 (HV0.01) and 50 g (HV0.05). The results of the microhardness measurements were used to determine the hardness increase Δ*HV* and the thickness of the layer hardened by ball burnishing *g_h_* (Figure 4).

Residual stress tests were carried out using a mechanical method [45]. Residual stresses in the surface layer of laser-cut specimens and specimens subjected to laser cutting followed by ball burnishing were calculated on the basis of specimen deformations that were revealed when the sequential layers of the material in which these stresses resided were removed. Layers were removed by chemical etching in a 4% nitric acid solution. Chemical etching was carried out on a special stand. The sample deformation value was measured with a digital sensor, the measuring tip of which was in contact with the surface of the etched sample. Measurement data were sent to a computer. Measurements of sample deflection were carried out with an accuracy of up to 0.0001 mm.The data from these tests were used to plot a graph showing the distribution of residual stress as a function of distance from the surface. This allowed us to determine the absolute value of compressive residual stresses *σ_max_* and the depth of compressive stresses *g_σ_* (Figure 5).

## 3. Results

### 3.1. Surface Roughness

Figure 5 shows the topography and a general view of a laser-cut surface. The surface formed in a laser cutting operation has numerous peaks and valleys (Figure 6a). Two zones with different levels of roughness are clearly visible in the images. In the zone where the laser beam entered the workpiece (A), the surface is covered with fine, evenly spaced straight drag lines. In the laser beam exit zone (B), the drag lines are curvilinear and deviate from standard slope (the intended path of the beam). During cutting, the layer of liquid metal in the exit zone becomes thicker and sticks to the bottom face and edge of the workpiece, creating a zone with large irregularities. This phenomenon is associated with an increase in the viscosity of the material being cut. As a result of uneven and rapid cooling of the molten metal, discontinuities and microcracks appear on the surface (Figure 6b,c).

The effect of burnishing force *F_n_* on the roughness parameters *Sa* and *Sz* is shown in Figure 7. There is a visible reduction in the roughness parameters *Sa* and *Sz* in relation to the values of these parameters after laser cutting. An increase in burnishing force *F_n_* in the range from 300 to 720 N caused a decrease in the surface roughness parameters *Sa* and *Sz*. This was due to the fact that as the burnishing force *F_n_* increased, the burnishing ball penetrated deeper into the workpiece, which resulted in more effective smoothing of asperities on laser-cut surfaces. The application of a burnishing force *F_n_* = 930 N resulted in only a slight increase in *Sa* and *Sz*, which may have been associated with excessive deformation of the surface of the workpiece.

Figure 8 shows the effect of path interval on surface roughness. An increase in path interval caused an increase in the values of *Sa* and *Sz*. At greater values of path interval *f_w_*, the tool paths are spaced farther apart from each other, which results in an uneven deformation of the striped structure and, ultimately, a surface with a greater roughness. The eight-fold increase in the path interval *f_w_* increased the Sa roughness parameter by 63%. On the other hand, during ball burnishing of the aluminium alloy EN AW-AlCu4MgSi (A), a two-fold increase in the path interval *f_w_* translated into an increase in the Sa parameter by 78% [8].

A burnishing ball with a smaller diameter *d_n_* makes contact with the workpiece over a smaller area, which allows high surface pressures to be achieved, resulting in more intensive levelling of asperities. As a result, a surface with a lower roughness is obtained (Figure 9).

In this study, the surface roughness parameters *Sa* and *Sz* were reduced by 21% to 60% relative to the values obtained after laser cutting. A similar reduction (40% to 94%) in surface roughness parameters was reported for 41Cr4 steel specimens subjected to ball burnishing [5]. The changes in the values of the roughness parameters obtained in the present experiment are also similar to those reported in [42]. Centrifugal shot peening of laser-cut C45 steel specimens allowed researchers to reduce the *Ra* parameter by 46% to 79% and the *Rz* parameter by 42% to 77% [42]. The finding that centrifugal shot peening led to larger changes in roughness parameters is associated with the fact that the burnishing elements used in the two processes impact the surface of the workpiece in different ways. During centrifugal shot peening, the balls hit the workpiece at a small angle, which, in addition to deformation, causes intensive shearing of asperity summits in the hard material [42]. By contrast, during ball burnishing, asperity summits are flattened out as the ball comes in contact with the processed surface.

A comparison of surface topographies as a function of the diameter of the burnishing tool shows that the asperities on the surface burnished with a *d_n_* = 8 mm ball are evenly deformed, even though the traces left by laser cutting have not been levelled out (Figure 10a). There is a visible reduction in the differences in asperities between the beam entry and beam exit zones. The summits of asperities were partially flattened out. When a ball with a diameter *d_n_* = 12 mm was used, the drag lines were slightly flattened (Figure 10b). The differences in surface roughness between the beam entry and beam exit zones are still visible. A comparison of the surfaces obtained using burnishing balls with different diameters (*d_n_* = 8 mm vs. *d_n_*= 12 mm) shows that the maximum height of the surface profile obtained with the 8 mm ball is about 32% lower than that obtained using the 12 mm ball. In the case of the surface burnished with the *d_n_* = 8 mm ball, the asperity peaks are more intensively reduced than the valleys, compared to the surface burnished with the *d_n_* = 12 mm ball. The largest height of the surface after burnishing with the *d_n_* = 8 mm ball decreased by 68% relative to the value obtained after laser cutting; burnishing with the *d_n_* = 12 mm ball reduced this parameter by 49%.

Figure 11, Figure 12 and Figure 13 show the effect of ball burnishing parameters on skewness *Ssk*. The changes in skewness *Ssk* as a function of the analysed burnishing parameters were similar to those for the roughness parameters *Sa* and *Sz*. An increase in burnishing force *F_n_* in the range from 300 to 720 N caused a “more complete” deformation of the stripes formed by laser cutting. The peaks of asperities were more strongly deformed than the valleys, which caused an increase in the absolute value of skewness (Figure 11). At *F_n_* = 930 N, a reduction in the absolute value of *Ssk* was obtained, which was most likely due to material flow.

At path intervals in the range *f_w_* = 0.05–0.28 mm, only very small changes in the value of *Ssk* were observed (Figure 12). When the path interval was greater than *f_w_* = 0.28 mm, there was no significant increase in the absolute value of *Ssk* in relation to the value observed for laser-cut parts. This can be explained by the fact that at large path intervals, the asperity summits do not undergo complete deformation. As a result, the surface of the processed material features spots that have not been burnished.

The use of variable ball diameters *d_n_* leads to small changes in *Ssk* as a function of this parameter, which fall within the range of standard deviation (Figure 13). The absolute values of *Ssk* of ball-burnished surfaces are several times higher than the values obtained for laser-cut parts, which is suggestive of an increase in the “functionality” of the processed surface. A surface with a negative skewness shows a greater capacity to transfer contact loads and a lower tribological wear in the presence of a lubricant [46].

Ball burnishing of laser-cut surfaces increased the absolute value of skewness from 90% to 820% in relation to the value obtained after laser cutting. In [14], the absolute value of Ssk was increased from 48% to 268% during the ball burnishing of milled X38CrMoV5-1 martensitic steel.

### 3.2. Microstructure and Microhardness

During laser cutting, the material being cut is strongly heated and then cooled down, which leads to the hardening of the material around the cut edge. The original ferrite-pearlite structure of C45 steel (Figure 14a) undergoes changes. The HAZ that forms around the cut edge has a different structure from the raw material. Martensite needles as well as ferrite and martensite are visible in the HAZ just next to the edge (Figure 14b). The laser cutting process was too short-lasting for ferrite grains to become enriched with carbon, which led to the formation of low-carbon martensite. Perlite, on the other hand, changed into high-carbon martensite.

In laser-cut specimens, the zone in which changes in microhardness occur is 0.3 mm wide (Figure 15). The microhardness increase Δ*HV* in the area of the cut edge is about 330 HV0.05, and it decreases away from the edge and deeper into the material, to reach core microhardness (Figure 15). Though the hardness and the thickness of the hardened layer increase only slightly as a result of ball burnishing (Figure 15), the increase is likely to translate into enhanced resistance to abrasive wear [22].

The ball-burnished surface layer constitutes a collection of plastically deformed crystals. This means that ball burnishing leads to structural changes and changes in the hardness of the material. Figure 16a shows the microstructure of laser-cut-and-ball-burnished material. In the edge zone with an HV 0.01 microhardness of 731.4 HV (Figure 16b), the martensite laths were broken up into smaller elements (fragmentation). As the tool comes in contact with the workpiece, phenomena occur that contribute to the defragmentation of the edge zone.

The hardness increase Δ*HV* and the thickness of the hardened layer *g_h_* depend on technological parameters. An increase in burnishing force *F_n_* leads to an increase in cold-work energy, which, when accumulated, causes an increase in the hardness of the areas close to the surface of the workpiece (Figure 17). The greatest microhardness increase was obtained when the burnishing force increased in the range from 300 to 510 N. At burnishing forces greater than *F_n_* = 510 N, the effect of this parameter on microhardness increase clearly decreased, which was associated with the occurrence of the state of saturation. The hardened layer was from 19 to 40 µm thick and its thickness increased with the increase in burnishing force, with the reservation that for forces in the range of 720–930 N, the increase in microhardness and thickness of the hardened layer was small.

When the burnishing tool is applied to the workpiece at larger intervals *f_w_*, the material becomes less structurally homogeneous. The channels carved by the ball are spaced farther apart from one another, which results in uneven deformation of the machined surface and ultimately translates into a smaller microhardness increase Δ*HV* and a decrease in the thickness of the hardened layer *g_h_* (Figure 18). The small hardness increase and the low thickness of the layer in which changes occur may be associated with the martensitic structure of the material, which is moderately susceptible to cold working.

Figure 19 shows the effect of ball diameter *d_n_* on the microhardness increase Δ*HV* and the thickness of the hardened layer *g_h_*. A maximum microhardness increase was obtained for the ball with the diameter *d_n_* = 8 mm, and the largest thickness of the hardened layer was for the ball with the diameter *d_n_* = 16 mm. For the investigated ball diameter range, changes in Δ*HV* and *g_h_* were small and fell within the range of standard deviation. Changes in Δ*HV* and thickness *g_h_* are associated with an increase in the radius of curvature of the ball, which increases with *d_n_*.

In the present study, the maximum hardness increment for laser-cut-and-ball-burnished C45 steel specimens was about 18%, which is similar to the results obtained in [4], where the degree of strain hardening was over 10% for steel with a hardness of 55 HRC and about 20% for steel with a hardness of 35 HRC.

### 3.3. Residual Stress

During laser cutting, the molten material re-solidifies as a result of the action of the laser beam, a process that takes place in unstable conditions. This gives rise to tensile stresses in the surface layer (Figure 20). In the specimens tested in the present study, these stresses resided at depths down to about 0.05 mm. Ball burnishing generates compressive residual stresses in the surface layer (Figure 20), whose value and depth depend on the technological parameters of the burnishing operation. The type of residual stresses generated in the surface layer of the workpiece is important from the point of view of the functional properties of the part, e.g., increased resistance to fatigue wear [22].

An increase in burnishing force *F_n_* caused an increase in the absolute value of compressive residual stress (Figure 21); the largest changes in residual stress were observed for forces in the range of *F_n_* = 510–720 N. When the ball was pressed against the workpiece with a greater force during a pass, the deeper-lying deformed zones elastically compressed the areas lying closer to the surface. The burnishing force had a smaller effect on the depth of accumulation of compressive residual stresses that lie at depths of 0.28–0.33 mm. The highest compressive stresses and the largest depth of accumulation of compressive residual stresses were obtained for *F_n_* = 930 N.

Figure 22 shows the effect of path interval *f_w_* on the maximum value of compressive residual stresses and the depth of their accumulation. An increase in *f_w_* means that burnishing marks are spaced farther apart from one another. The surface being processed is deformed to a lesser degree, which results in a decrease in the absolute maximum value of compressive residual stresses in the surface layer of the ball-burnished part. The depth of residual stresses does not change significantly for path intervals in the range *f_w_* = 0.05–0.17 mm. At higher path intervals, there is a visible decrease in the depth of penetration of residual stresses.

As the diameter of the ball increases, the absolute value of maximum residual stresses *σ_max_* decreases, which can be explained by the increase in the radius of curvature of the ball and a decrease in pressure per unit area. At the same time, the depth of penetration of compressive residual stresses *g_σ_* increases (Figure 23). When the 16 mm diameter ball was used, the depth of penetration of compressive residual stresses was 20% greater and the value of those stresses was 17% smaller than for the *d_n_* = 8 mm ball.

The use of ball burnishing as a finishing treatment for laser-cut parts led to the formation of compressive residual stresses in the surface layer of the workpiece. Grinding, which has so far been used as a standard procedure for finishing laser-cut parts, generates tensile stresses in the surface layer of the material [44]. The maximum values of compressive residual stress *σ_max_* obtained in the present experiments were lower than those obtained by centrifugal shot peening, for which |*σ_max_*| = 448–737 MPa; however, the depth of compressive residual stresses was similar to that obtained in centrifugal shot peening tests [42].

## 4. Conclusions

The results of the ball burnishing experiments conducted using laser-cut C45 steel parts lead to the following conclusions:Ball burnishing of laser-cut C45 steel parts reduced the surface roughness parameters *Sa* and *Sz* by 21% and 60%, respectively, in relation to the values obtained by laser cutting. The change parameters *Sa* and *Sz* becoming more prominent for the variable *F_n_*.Ball burnishing leads to the reorganization of the geometric structure of the surface. As a result of burnishing, the absolute value of skewness *Ssk* increased by 90% to 820% relative to the value obtained by laser cutting.Ball burnishing of laser-cut parts caused structural changes in the material. Martensite laths were broken up into smaller fragments (fragmentation).The maximum microhardness increment obtained during ball burnishing was from Δ*HV* = 29 HV0.05 to Δ*HV* = 95 HV0.05, and the depth of the hardened layer *g_h_* ranged from 16 to 40 µm. The microhardness of the surface layer and the thickness of the hardened layer *g_h_* increased with an increase in burnishing force *F_n_*. An increase in ball diameter caused a slight increase in *g_h_*.Ball burnishing generated compressive residual stresses in the surface layer of laser-cut parts, whose absolute maximum value ranged from 257 to 380 MPa. The stresses resided at a depth of 0.25–0.40 mm, depending on the technological parameters of the burnishing process. As the burnishing force *F_n_* increased, the absolute value of compressive residual stress *σ_max_* and the depth of compressive stresses *g_σ_*. On the other hand, the use of a ball with a larger diameter caused a decrease in the value of compressive residual stress *σ_max_*

## Figures and Tables

**Figure 1 materials-13-03429-f001:**
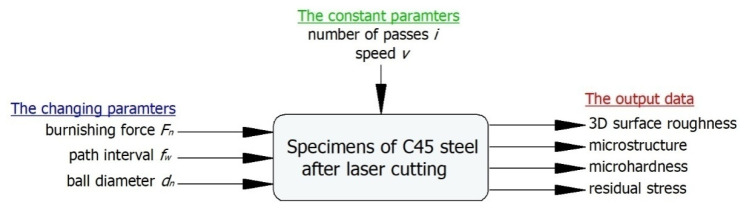
Plan on the experiment.

**Figure 2 materials-13-03429-f002:**
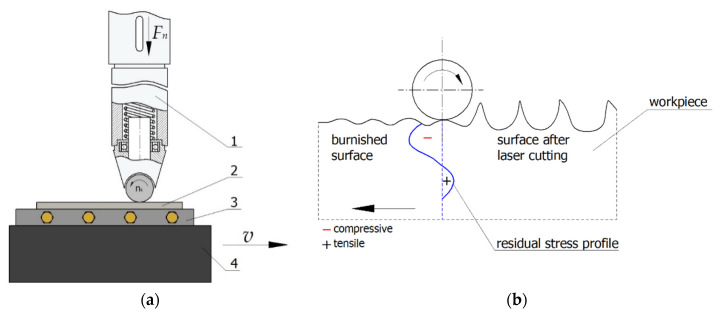
Schema of ball burnishing process of flat surface on vertical machining centre: (**a**) ball burnishing kinematics, (**b**) principle of ball burnishing of the surface after laser cutting (*v*—speed, *F_n_*—burnishing force) (1—burnishing tool, 2—sample, 3—holder, 4—vice).

**Figure 3 materials-13-03429-f003:**
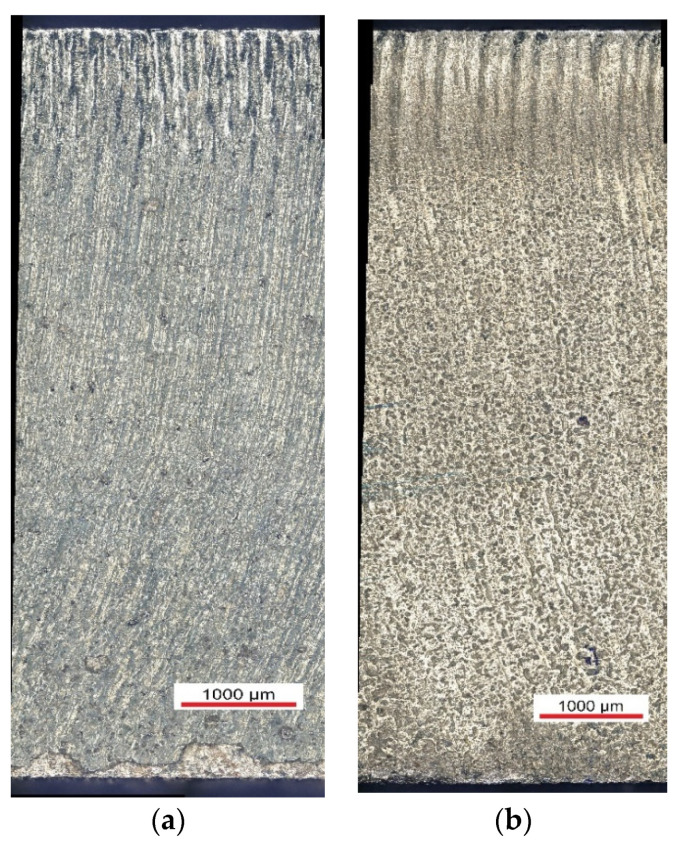
The surface after laser cutting (**a**) and laser cutting and ball burnishing (**b**) (*F_n_* = 720 N, *f_w_* = 0.05 mm *d_n_* = 8 mm).

**Figure 4 materials-13-03429-f004:**
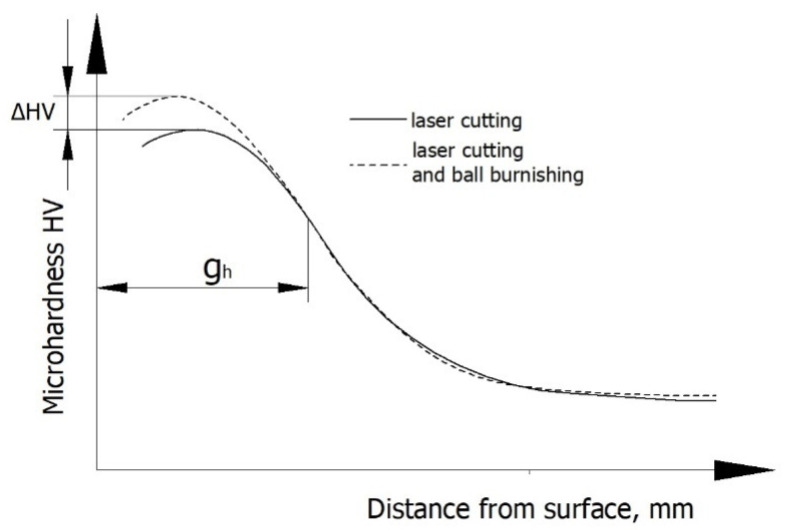
Methodology for determining the increased microhardness Δ*HV* and thickness of the hardened layer *g_h_*.

**Figure 5 materials-13-03429-f005:**
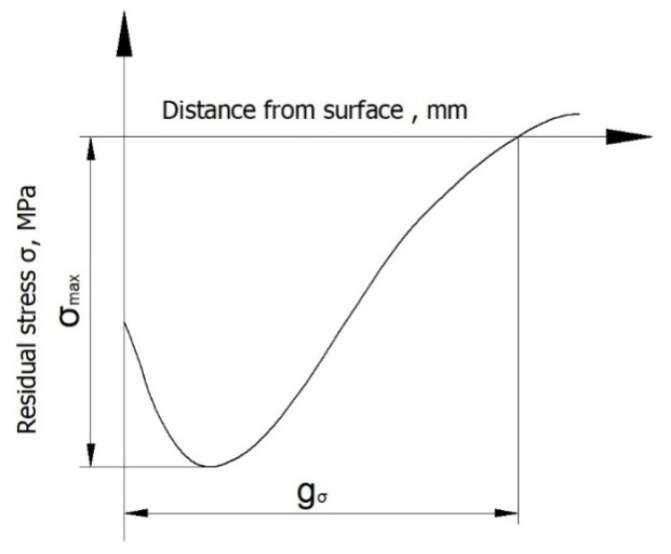
Methodology for determining the absolute value of compressive residual stresses *σ_max_* and the depth of compressive stresses *g_σ_*.

**Figure 6 materials-13-03429-f006:**
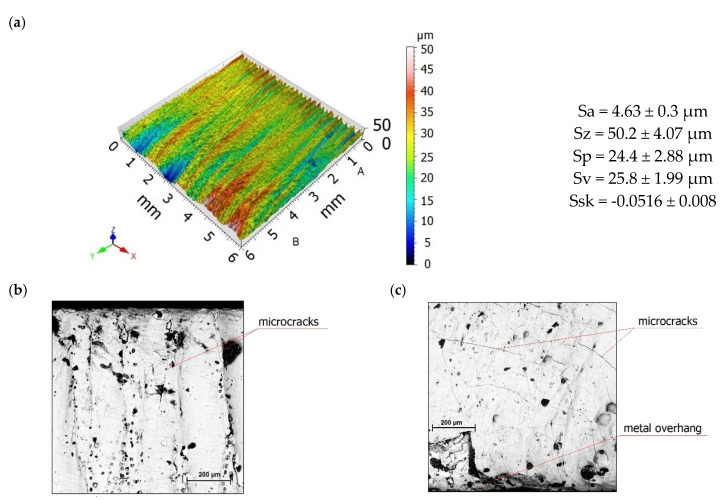
Surface topography (**a**) and scanning micrographs of the surface after laser cutting in the entrance zone (**b**) and in the exit zone (**c**).

**Figure 7 materials-13-03429-f007:**
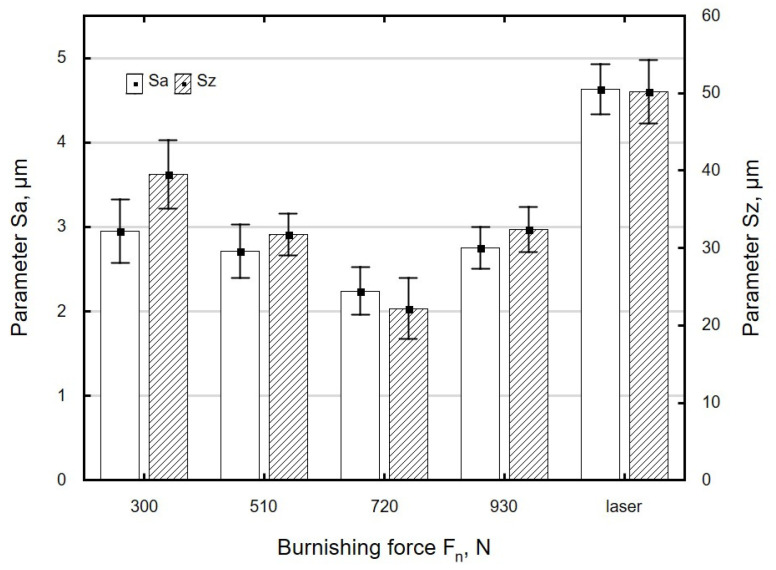
The effect of burnishing force *F_n_* on the roughness parameter *Sa* and *Sz* specimens after laser cutting and ball burnishing (*f_w_* = 0.17 mm, *d_n_* = 8 mm).

**Figure 8 materials-13-03429-f008:**
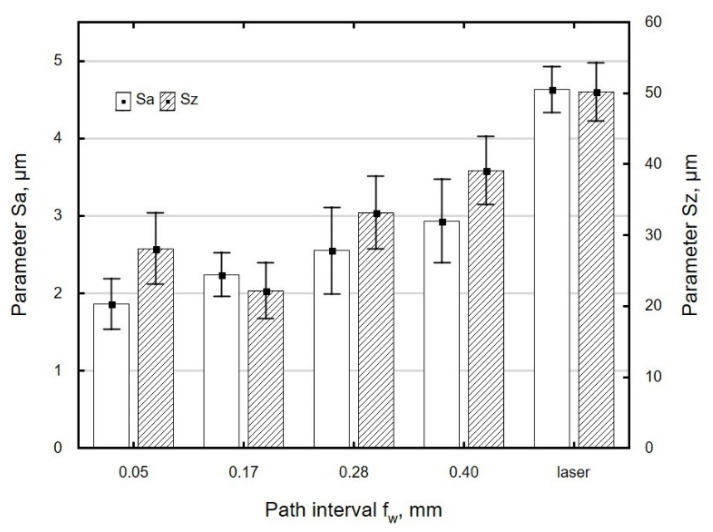
The effect of path interval *f_w_* on the roughness parameter *Sa* and *Sz* specimens after laser cutting and ball burnishing (*F_n_*= 720 N, *d_n_* = 8 mm).

**Figure 9 materials-13-03429-f009:**
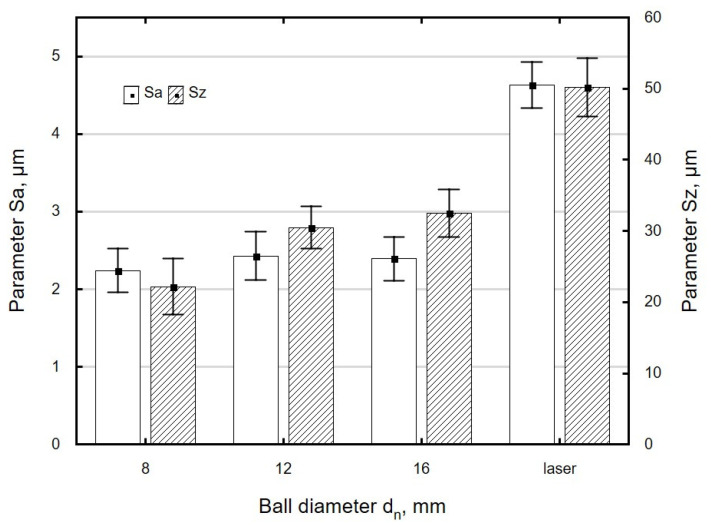
The effect of ball diameter *d_n_* on the roughness parameter *Sa* and *Sz* specimens after laser cutting and ball burnishing (*F_n_* = 720 N, *f_w_* = 0.17 mm).

**Figure 10 materials-13-03429-f010:**
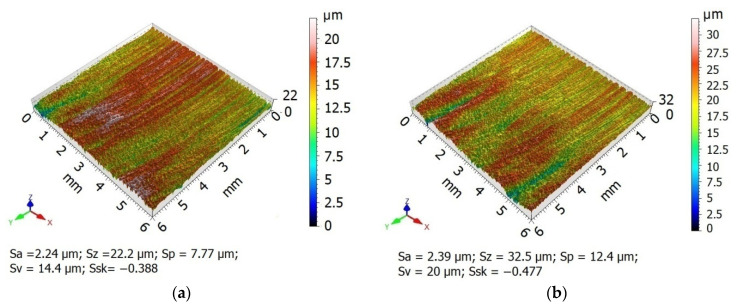
Surface topographies specimens after laser cutting and ball burnishing: (**a**) *d_n_* = 8 mm, (**b**) *d_n_* = 12 mm (*F_n_* = 720 N, *f_w_* = 0.17 mm).

**Figure 11 materials-13-03429-f011:**
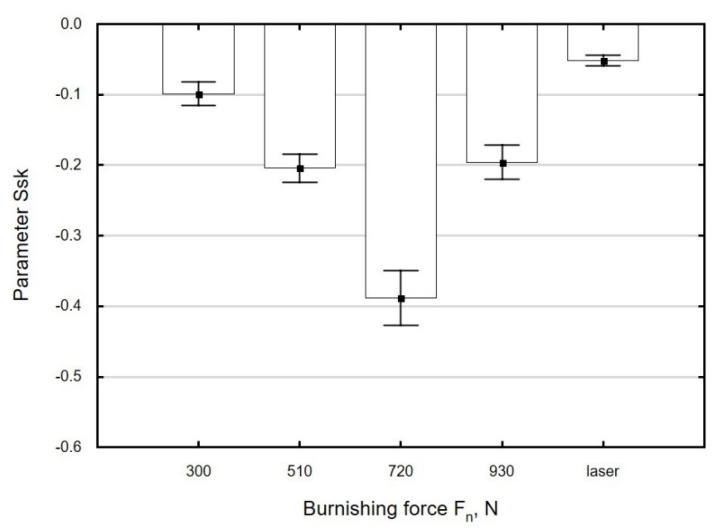
Effect of burnishing force *F_n_* on the parameter *Ssk* of specimens after laser cutting and ball burnishing (*f_w_* = 0.17 mm, *d_n_* = 8 mm).

**Figure 12 materials-13-03429-f012:**
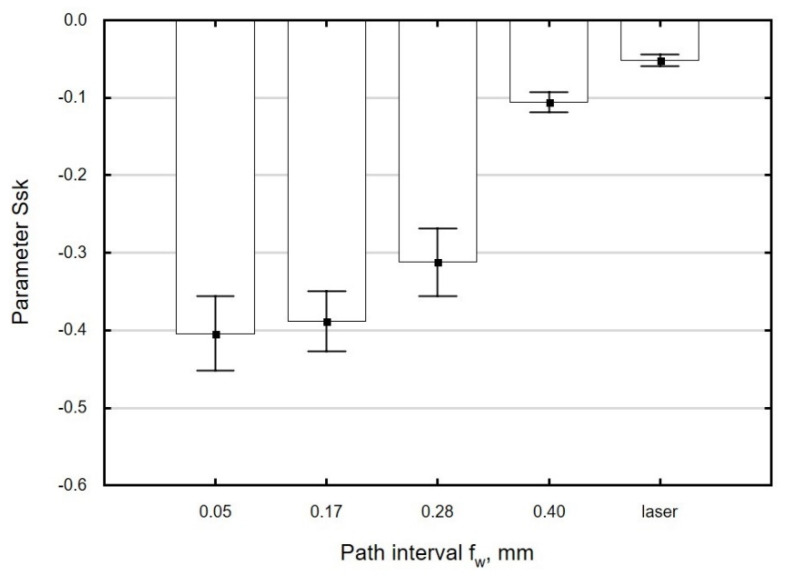
Effect of path interval *f_w_* on the parameter *Ssk* of specimens after laser cutting and ball burnishing(*F_n_* = 720 N, *d_n_* = 8 mm).

**Figure 13 materials-13-03429-f013:**
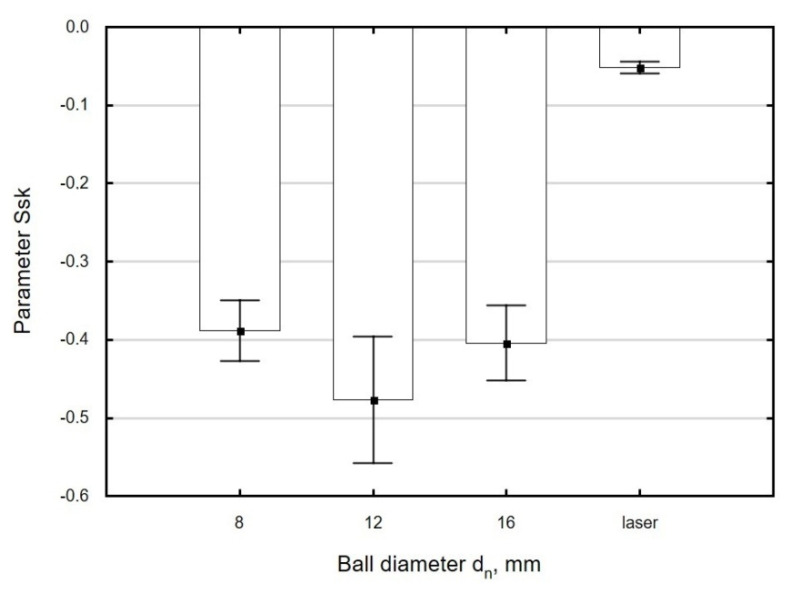
Effect of ball diameter *d_n_* on the parameter *Ssk* of specimens after laser cutting and ball burnishing (*F_n_* = 720 N, *f_w_* = 0.17 mm).

**Figure 14 materials-13-03429-f014:**
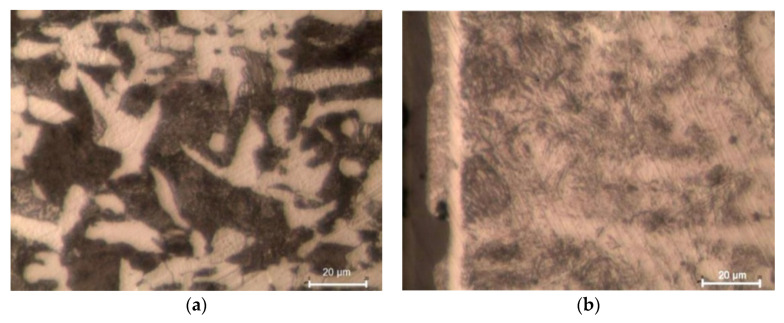
Microstructure of C45 steel: (**a**) before laser cutting, (**b**) heat affected zone (HAZ) after laser cutting (light optical microscopy, etched by 2% Nital).

**Figure 15 materials-13-03429-f015:**
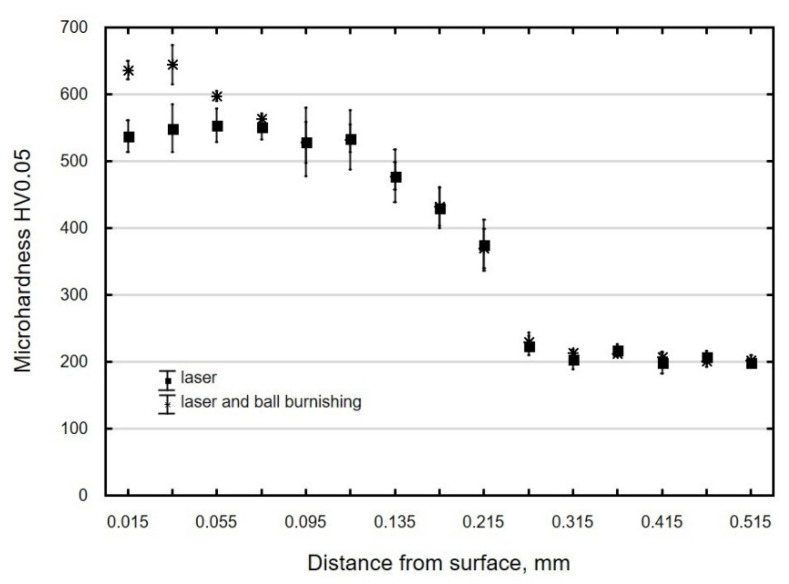
Microhardness distribution of C45 steel element surface layer after laser cutting and laser cutting and ball burnishing (*F_n_* = 720 N, *f_w_* = 0.17 mm, *d_n_* = 8 mm).

**Figure 16 materials-13-03429-f016:**
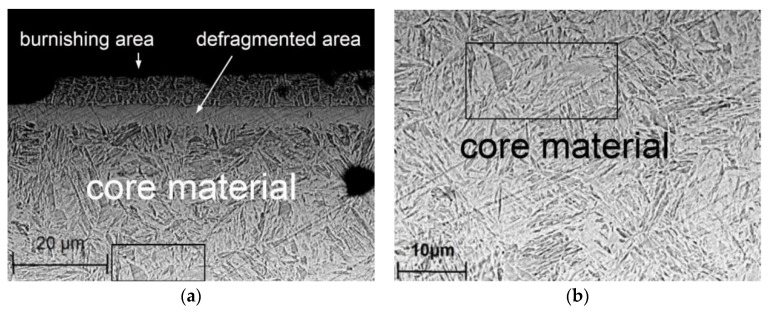
Microstructure steel C45 after laser cutting and ball burnishing (*F_n_* = 720 N, *f_w_* = 0.17 mm, *d_n_* = 8 mm): (**a**) burnishing area and core material, (**b**) core material (scanning microscope PhenonProX).

**Figure 17 materials-13-03429-f017:**
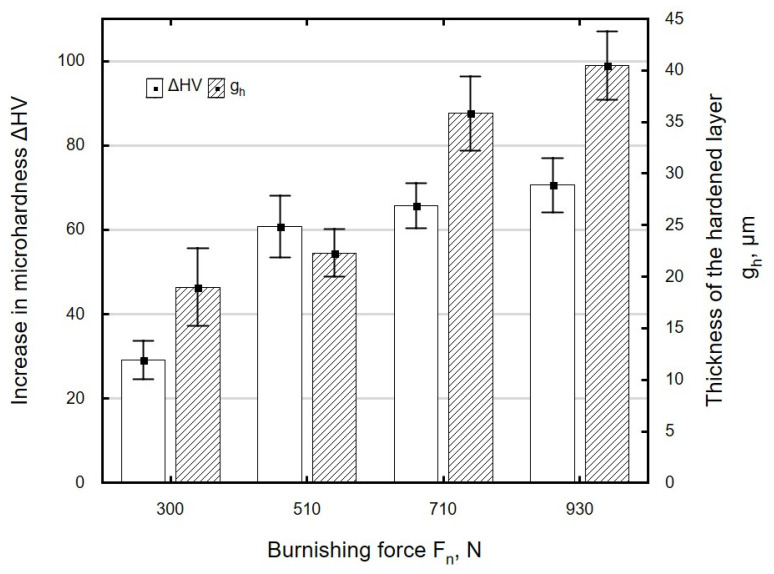
Influence of burnishing force *F_n_* on microhardness increase Δ*HV* and thickness of the hardened layer *g_h_* of specimens after laser cutting and ball burnishing (*f_w_* = 0.17 mm, *d_n_* = 8 mm).

**Figure 18 materials-13-03429-f018:**
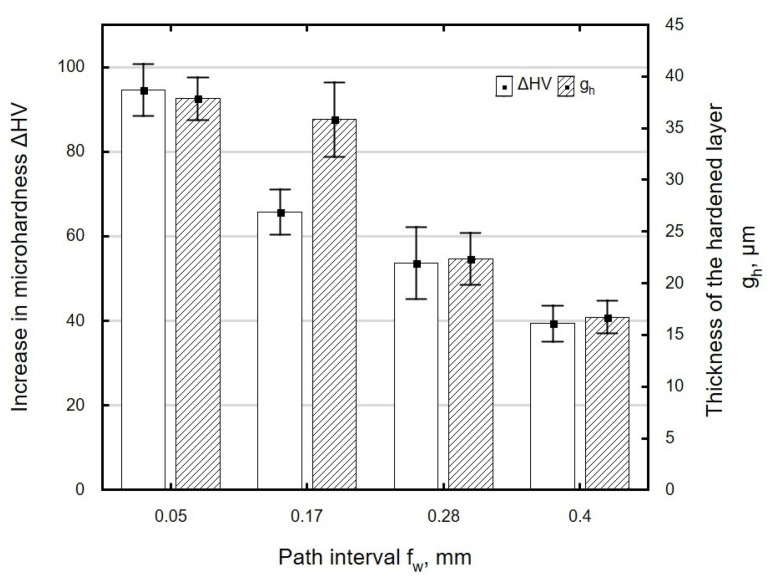
Influence of path interval *f_w_* on microhardness increase Δ*HV* and thickness of the hardened layer *g_h_* of specimens after laser cutting and ball burnishing (*F_n_* = 720 N, *d_n_* = 8 mm).

**Figure 19 materials-13-03429-f019:**
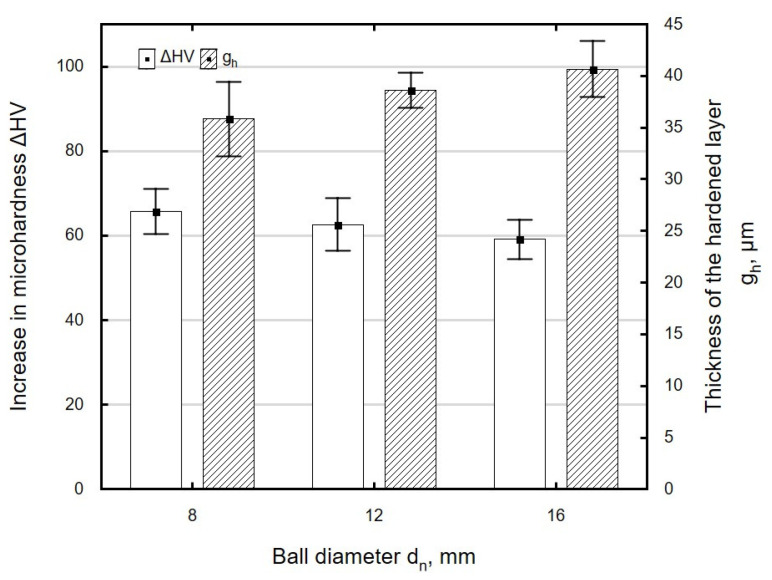
Influence of ball diameter *d_n_* on microhardness increase Δ*HV* and thickness of hardened layer *g_h_* specimens after laser cutting and ball burnishing (*F_n_* = 720 N, *f_w_* = 0.17 mm).

**Figure 20 materials-13-03429-f020:**
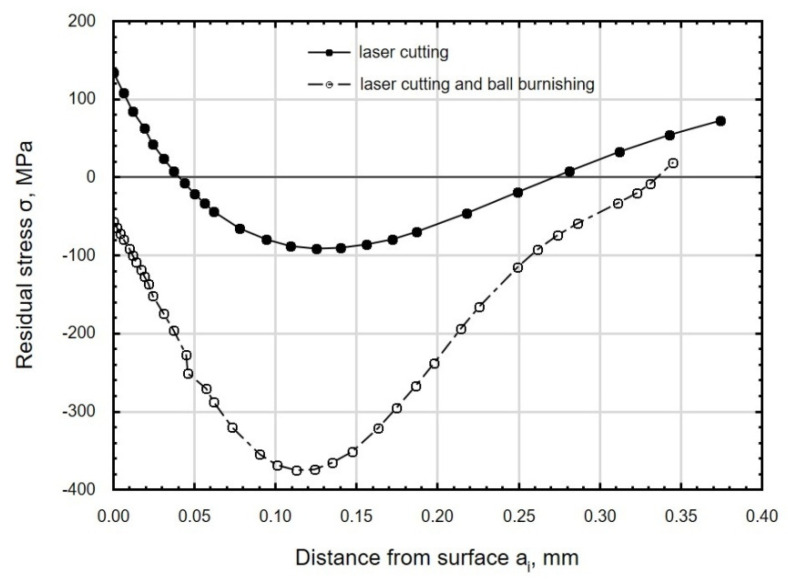
Distribution of residual stress in the function distance from C45 steel sample after laser cutting and ball burnishing (*F_n_* = 720 N, *f_w_* = 0.17 mm, *d_n_*= 8 mm).

**Figure 21 materials-13-03429-f021:**
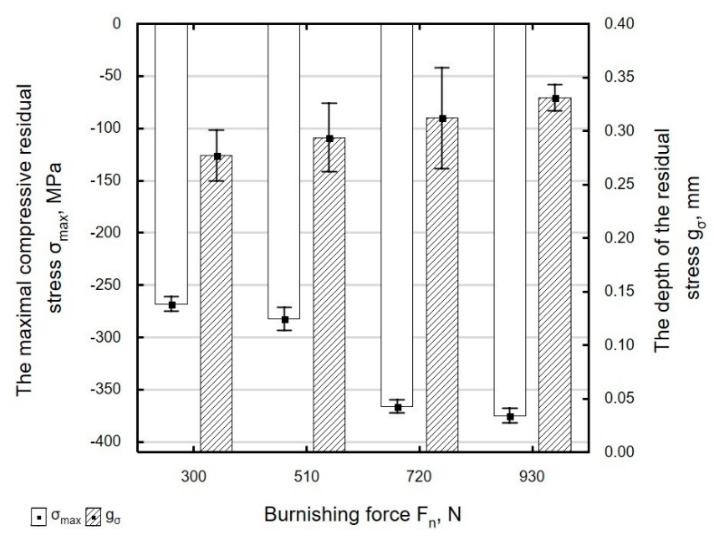
Influence of burnishing force *F_n_* on the value of maximal residual stress *σ_max_* and the depth of the accumulation compressive residual stress *g_σ_* of laser cut and ball burnishing specimens (*f_w_* = 0.17 mm, *d_n_* = 8 mm).

**Figure 22 materials-13-03429-f022:**
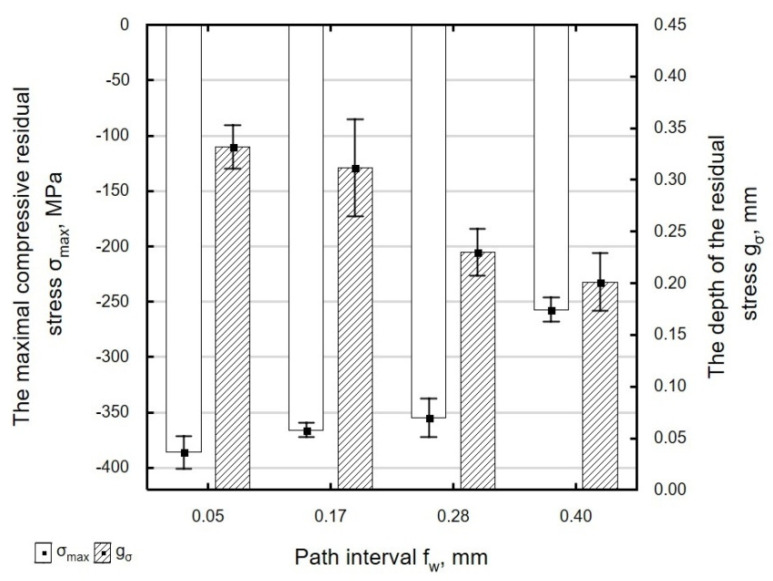
Influence of path interval *f_w_* on the value of maximal residual stress *σ_max_* and the depth of the accumulation compressive residual stress *g_σ_* of laser cut and ball burnishing specimens (*F_n_* = 720 N, *d_n_* = 8 mm).

**Figure 23 materials-13-03429-f023:**
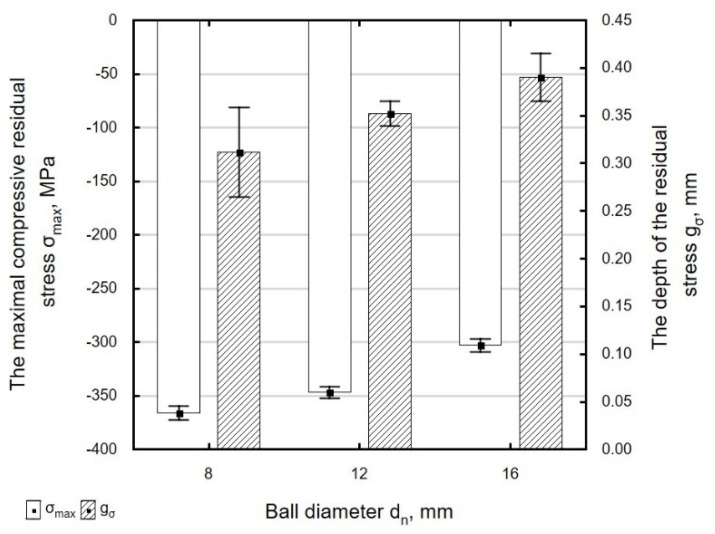
Influence of ball diameter *d_n_* on the value of maximal residual stress *σ_max_* and the depth of the accumulation compressive residual stress *g_σ_* of laser cut and ball burnishing specimens (*F_n_* = 720 N, *f_w_* = 0.17 mm).

**Table 1 materials-13-03429-t001:** Ball burnishing conditions.

Burnishing Force *F_n_*, N	Path Interval *f_w_*, mm	Ball Diameter *d_n_*, mm
300	0.17	8
510		
720		
930		
720	0.05	
	0.28	
	0.40	
	0.17	12
		16
Speed *v* = 2500 mm/min		
Number of passes *i* = 1		

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
