# Peer review of "Selected Properties of the Surface Layer of C45 Steel Parts Subjected to Laser Cutting and Ball Burnishing"

_materials, 2020, doi:10.3390/ma13153429_

Round 1
Reviewer 1 Report
The authors present the influence of ball burnishing parameters on the roughness, microhardness,residual microhardness of the surface layer of laser-cut C45 steel parts. The topic is worth of investigation and the results are relevant for the field. However, in this manuscript the introduction has to be improved in order to become interesting to the readership. Some problems were given as below:
- There are too many chapters to describe “Ball burnishing” method in Introduction part, the authors should describe it briefly.
- The authors should make a table about the experiment parameters (Line 191- Line 196).
- What is the test method of residual stress in this text and what is the accuracy? Please explain on this. (Line 219-Line 226)
- Why the depth of penetration of compressive residual stresses increases with the diameter of the ball increases? Because the diameter of the ball increases, as we all know, the pressure decrease. (Line 440)
- Some minor mistakes need to be corrected, such as Figures 10 ÷ 12 (Line 304)should be corrected as “Figures 10-12”, (Fig. 10) (Line 347) should be corrected as “Figure 14”, etc..
- The Conclusions are too tedious. It should be brief and clear. Write those matters that you want the reader to have learned from your work.
Author Response
The answer to the review is attached. Yours sincerely Agnieszka Skoczylas

Reviewer 2 Report
The work related to laser cutting and ball burnishing process is interesting and lots of experimental tests including surface roughness, hardness, residual stress and microstructure have been done, and the corresponding analysis of the tested results have been discussed in details.
I would like to give two suggestions for improving the quality.
1. Please show some pictures of the samples to explain the surface after laser cutting and the surface after ball burnish.
2. Please give more detailed information about the residual stress measurement.
Author Response

(The authors gave the same response as above.)

Reviewer 3 Report
The paper, dealing with properties of the surface layer of the C45 steel subjected to ball burnishing after laser cutting, is overall well written. The introduction is quite extensive and and provides a good background to the reader, there are sufficient (44) references to the literature. The study is systematic and the results are relatively clearly presented. However, there are points that should be improved. I recommend the paper for publication with minor, but mandatory revision, especially of figures concerning the microstructure of the steel.
Comments:
line 50: Ball burnishing of increased hardness steel .. . is not clear. Does it mean "Ball burnishing of steel with increased hardness?" Please clarify
line 203: I suggest to replace the word "device" by "mechanical roughness tester". It is more precise and it leads a non-specialist reader directly to the method used (it cannot be confused with laser profilometer)
line 208: there is a misprint in Phenom ProX scanning microscope (in the paper there is Phenon). In the first reading I suggested to add "electron" microscope to be more precise and clear for the reader. However, after reading the whole paper I found that there is not any electron microscope micrograph presented, so the part of the sentence: "and a Phenom ProX scanning microscope (Thermo Fisher Scientific)." should be omitted.
Line 241: Numbers and letters in Fig. 5a are too small to be clearly visible in the printed version of the paper. Values of 0.5, 1.5 etc. could be omitted and the font of numbers could be enlarged, as well as "mm". Letters A and B have no reference in the text neither in the figure caption and they should be omitted. Font size for "macrocracks" and "metal overhang" should be increased. The figure caption should be modified: Figure 5: Surface topography (a) and light micrographs of the ...
Line 301: Fig. 9 - font of the numbers and letters should be enlarged in the same manner as for Fig. 5a
Line 343: Brightness and contrast of the micrographs should be improved (see attached file for the result). The caption should be completed e.g.: Figure 13. Microstructure of C45 steel: (a) before laser cutting, (b) HAZ afer laser cutting (light optical microscopy, etched by ....).
Line 360: Brightness of Fig. 15a should be increased (see attached file for the result). Furthermore, the Fig. 15b is not correct. In the paper figure caption there is “(b) burnishing area”, but it is not true, as follows from the microstructural feature in the framed region in the core material of Fig. 15a, which is the same in Fig. 15b (attachment). I suggest to improve Fig. 15a by direct marking of the burnished and defragmented areas and the core material. Fig. 15b should be replaced or omitted. Furthermore, it should be written in the caption that it is a light optical micrograph, and the etchant should be also indicated (2% Nital?).

Author Response

(The authors gave the same response as above.)

Reviewer 4 Report
The paper is about the burnishing process of C45 Steel parts. The introduction reflects a good study of the state of the art on the subject. Although the burnishing process on this material has been extensively discussed, the paper explores its influence on laser cut parts, which is certainly novel. In general, it is very well written and what it wants to convey is well understood.
There are a few things I would like to clarify, before recommending it for final publication.
- In Figure 1, there is an extra "m" in the word number
- What did you take into account to select the values of the parameters used? For example: Fn= 300-930 N, fw= 0.05-0.4 mm, dn= 8-16 mm. It is necessary to explain the reason. Also to select v= 2500 mm/min and i=1 pass.
- In figure 2, the font used in the variables, for example, V and Fn; has to be the same than those used in the figure caption. If No, they seem like two different variables.
- Why the scan area was selected as 6 x 6 mm for the 3D topography measurement?
- Some results can be discussed more according to the literature review. For example, the influence of path interval increase in the values of Sa and Sz.
- In Figure 7, the authors show the effect of path interval on the roughness parameters for Fn= 720 N and dn= 8 mm. Do the results are comparable to the rest of the conditions? I mean, for the other forces and ball diameter? The same thing happens in the rest of the figures that explain this type of results.
- The majority of the figure show graphs for the conditions mentioned before. Probably these are the best conditions found. Is that so?
- According to figure 8, the best results were obtained in case of 8-mm ball diameter. Other studies talk about that. What did the rest of the authors say about that?
- In figure 19, laser cutting and ball burnishing… “and” it is misspelt.
- Dis the authors used any technique of DoE to validate the statistical significance of the results? How many specimens were used to measure each property in each condition evaluated? Is there any interaction between the selected parameters?
Author Response

(The authors gave the same response as above.)
